

# A Cdk1 phosphomimic mutant of MCAK impairs microtubule end recognition

Hannah R. Belsham and Claire T. Friel

School of Life Sciences, University of Nottingham, Nottingham, United Kingdom

## ABSTRACT

The microtubule depolymerising kinesin-13, MCAK, is phosphorylated at residue T537 by Cdk1. This is the only known phosphorylation site within MCAK's motor domain. To understand the impact of phosphorylation by Cdk1 on microtubule depolymerisation activity, we have investigated the molecular mechanism of the phosphomimic mutant T537E. This mutant significantly impairs microtubule depolymerisation activity and when transfected into cells causes metaphase arrest and misaligned chromosomes. We show that the molecular mechanism underlying the reduced depolymerisation activity of this phosphomimic mutant is an inability to recognise the microtubule end. The microtubule-end residence time is reduced relative to wild-type MCAK, whereas the lattice residence time is unchanged by the phosphomimic mutation. Further, the microtubule-end specific stimulation of ADP dissociation, characteristic of MCAK, is abolished by this mutation. Our data shows that T537E is unable to distinguish between the microtubule end and the microtubule lattice.

## INTRODUCTION

Mitotic Centromere Associated Kinesin (MCAK) is a member of the kinesin-13 family of microtubule depolymerising kinesins. MCAK plays crucial roles in the cell cycle both in building the mitotic spindle and in correcting erroneous microtubule-kinetochore attachments. Therefore, both the localisation and depolymerisation activity of MCAK must be tightly regulated throughout the cell cycle. This regulation is primarily achieved through phosphorylation.

MCAK is regulated through the action of various mitotic kinases, including the aurora kinases, polo like kinase 1 and p21 activated kinase 1 (*Andrews et al., 2004*; *Lan et al., 2004*; *Pakala et al., 2012*; *Zhang et al., 2011*; *Zhang, Ems-McClung & Walczak, 2008*). These kinases phosphorylate MCAK at various sites within the N and C-terminal domains and in the neck region. Only one phosphorylation site within MCAK's motor domain has been identified to date. Threonine 537, which is located adjacent to the α4 helix on the tubulin-binding face of the motor domain (Fig. 1A), is phosphorylated by the cyclin-dependant kinase Cdk1 (*Sanhaji et al., 2010*). A phosphomimic mutant at this position, T537E, has reduced depolymerisation activity and overexpression of this mutant in cells leads to misaligned chromosomes, metaphase arrest and reduced intercentromeric distances (*Sanhaji et al., 2010*).

Corresponding author
Claire T. Friel,
claire.friel@nottingham.ac.uk

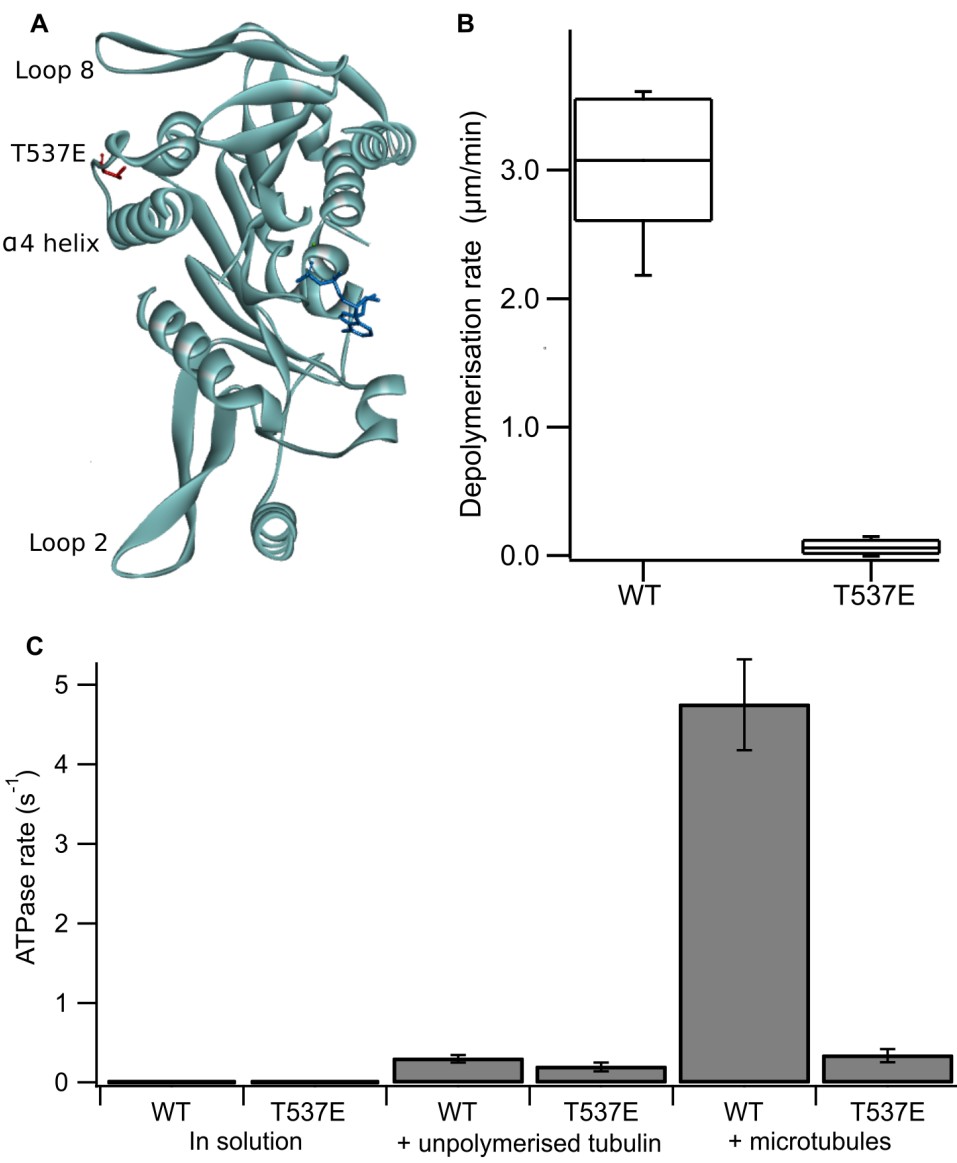

**Figure 1** **The phosphomimic mutant T537E has reduced depolymerisation activity and reduced microtubule-stimulated ATPase activity.** (A) Location of T537 in the crystal structure of the MCAK motor domain (PDB ID: 2HEH). T537 (red) is located adjacent to the $\alpha$4 helix. The $\alpha$4 helix, loop 2 and loop 8 are at the microtubule binding interface of MCAK. The nucleotide is shown in dark blue. (B) Depolymerisation rate of WT MCAK and T537E mutant. Distribution of depolymerisation rates of single microtubules after the addition of 40 nM MCAK. The box represents the central 50% of the distribution, the central line the median and the whiskers the 10th–90th percentile. (C) ATPase rate of WT MCAK and T537E mutant in solution, in the presence of 10 $\mu$M unpolymerised tubulin and in the presence of 10 $\mu$M microtubules. Error bars are ±standard deviation.

MCAK is a specialist microtubule depolymerising kinesin and its ATP turnover cycle is tailored to this function. While many kinesins have a translocating, "walking" action along the microtubule, MCAK diffuses on the microtubule lattice in the ADP-bound state. The microtubule end specifically accelerates ADP dissociation, stimulating exchange of ADP

for ATP, which allows MCAK to bind tightly at the end of the microtubule. The binding of ATP.MCAK at the microtubule end promotes dissociation of tubulin dimers and thereby microtubule depolymerisation (*Friel & Howard, 2011*).

To understand how phosphorylation by Cdk1 impairs MCAK's depolymerisation activity we studied the effect of the phosphomimic substitution T537E on the molecular mechanism of microtubule depolymerisation. Here we show that the phosphomimic mutant, T537E, cannot distinguish between the microtubule end and the microtubule lattice, an ability characteristic of wild-type MCAK and other kinesins which regulate microtubule dynamics.

## METHODS

### Proteins
Full length human MCAK-his6 and MCAK-his6-EGFP in wild type and mutated forms were expressed in Sf9 cells (Invitrogen, Carlsbad, CA, USA) and purified using nickel affinity chromatography as described previously (*Helenius et al., 2006*). MCAK concentrations are given as monomer concentrations. Porcine brain tubulin was purified as described previously (*Castoldi & Popov, 2003*). Single cycled, fluorescently labelled microtubules and double cycled microtubules were prepared as described previously (*Patel et al., 2016*). The concentration of microtubules is given as the concentration of polymerised tubulin.

### Microtubule depolymerisation
Microtubule depolymerisation rates were determined by measuring the length of immobilised, GMPCPP-stabilised, rhodamine labelled, single cycled microtubules over time after the addition of 40 nM MCAK as described previously (*Patel et al., 2016*).

### ATPase assays
ATPase rates in solution were measured using 3 $\mu$M MCAK and monitoring the production of ADP by HPLC as described previously (*Friel, Bagshaw & Howard, 2011*; *Friel & Howard, 2011*; *Patel et al., 2016*). For assays with tubulin or double-cycled microtubules 0.1 $\mu$M MCAK was used and the production of ADP was monitored by linking it to the oxidation of NADH (*De La Cruz & Ostap, 2009*). For both assays the change in concentration of ADP per second was divided by the concentration of MCAK to give the ATPase activity per second per motor domain.

### Single molecule TIRF assays
Single molecules of MCAK-GFP were observed on immobilised, GMPCPP-stabilised, rhodamine labelled, single cycled, microtubules using TIRF microscopy as described previously (*Patel et al., 2016*). Kymographs for individual microtubules were used to measure the time individual MCAK-GFP molecules spent at the microtubule end and on the lattice.

### ADP dissociation assays
The dissociation of mantADP from MCAK was measured as described previously (*Patel et al., 2014*), in solution or with the addition of 10 $\mu$M tubulin or 5.7 $\mu$M double cycled

microtubules (chosen to have a comparable number of microtubule ends as the ATPase assay with microtubules). The fluorescence traces were fitted to a single exponential, or double exponential if required, with an additional linear function to account for the photobleaching of mant.

## RESULTS

### The phosphomimic mutant T537E reduces depolymerisation activity and microtubule-stimulated ATPase activity

Firstly, we confirmed the effect of the substitution T537E on MCAK's depolymerisation activity *in vitro*. It has been shown previously at high concentration (500 nM) that the mutation T537E decreases depolymerisation activity 4-fold relative to wild-type MCAK (*Sanhaji et al., 2010*). We measured depolymerisation activity at 40 nM, the concentration at which the fastest microtubule depolymerisation for wild-type MCAK is observed (*Helenius et al., 2006*). Under these conditions the phosphomimic mutant had a depolymerisation rate 50-fold slower than the wild-type ($0.06 \pm 0.06\,\mu m\,min^{-1}$ and $3.04 \pm 0.53\,\mu m\,min^{-1}$ (mean $\pm$ standard deviation), respectively) (Fig. 1B). Thus, confirming that the phosphomimic substitution dramatically decreases MCAK's depolymerisation activity.

We next measured the ATPase activity of T537E in the absence of tubulin, in the presence of unpolymerized tubulin and in the presence of microtubules. The ATPase rate of the T537E mutant in solution was not significantly different to wild type ($5.33 \pm 0.33 \times 10^{-3}\,s^{-1}$ compared with $4.47 \times 10^{-3} \pm 2.60 \times 10^{-3}\,s^{-1}$, $p = 0.6002$) (Fig. 1C). Similarly, the ATPase in the presence of unpolymerized tubulin was not significantly affected by the phosphomimic mutation: $0.194 \pm 0.055\,s^{-1}$ compared with $0.299 \pm 0.047\,s^{-1}$ for wild-type MCAK, $p = 0.0658$ (Fig. 1C). These data indicate that the phosphomimic mutant folds correctly as it turns over ATP at a similar rate to wild-type both in solution and in the presence of unpolymerized tubulin. These data also suggest that T537E can still interact with unpolymerized tubulin, as its ATPase rate is accelerated by unpolymerized tubulin to the same degree as wild type. By contrast, there is a dramatic difference in the microtubule-stimulated ATPase activity of T537E relative to wild-type MCAK. The microtubule-stimulated ATPase rate of T537E is 14-fold slower than wild type MCAK: $0.335 \pm 0.081\,s^{-1}$ and $4.75 \pm 0.055\,s^{-1}$, respectively (Fig. 1C). The ATPase rate for T537E was in fact similar to the ATPase for both wild-type and T537E in the presence of unpolymerized tubulin. The difference between the ATPase rates in the presence of unpolymerized tubulin compared to the presence of microtubules for wild-type MCAK has previously been shown to be due to the acceleration of ADP dissociation caused specifically by microtubule ends (*Friel & Howard, 2011*). Therefore, the reduction in the ATPase of T537E to a similar level to that in the presence of unpolymerized tubulin suggests this mutation may have specifically impaired MCAK's ability to recognise the microtubule end.

### The mutation T537E abolishes long microtubule end residence events

The ATPase activity of wild-type MCAK is maximally accelerated by microtubule ends (*Friel & Howard, 2011*; *Hunter et al., 2003*). The residue T537 is near the α4 helix of

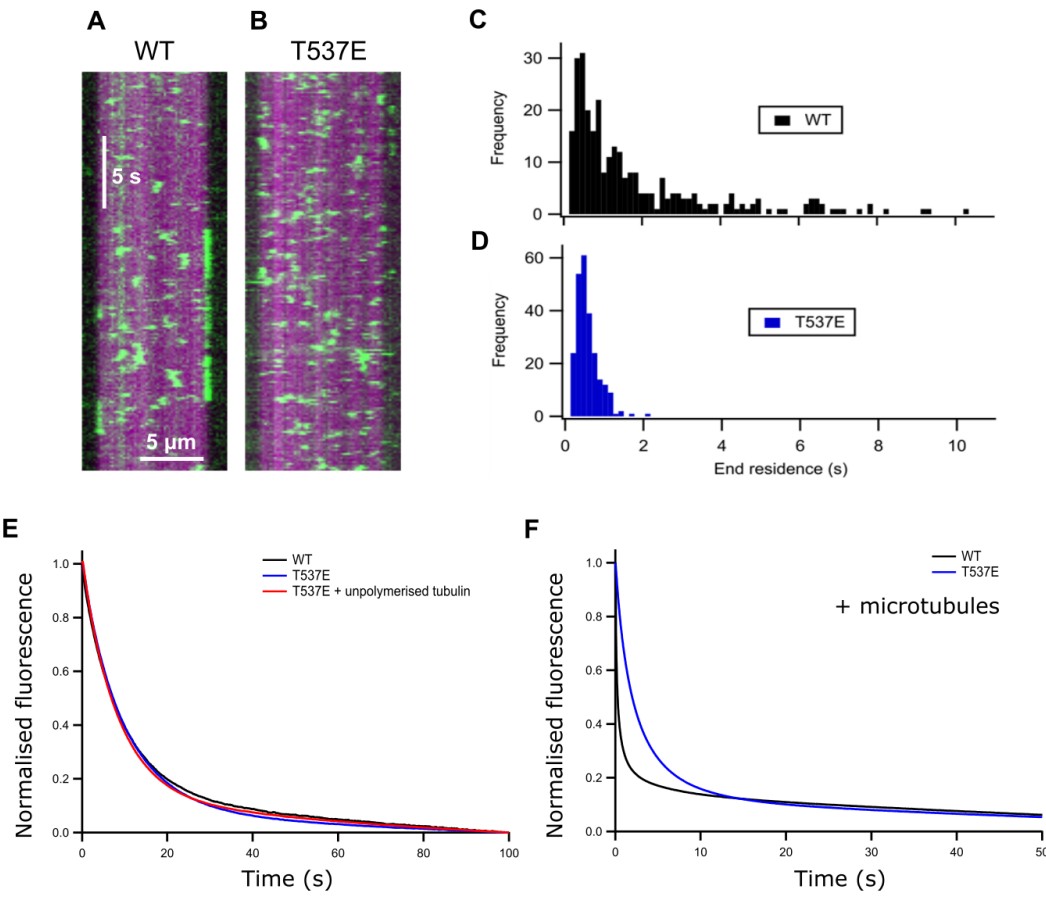

**Figure 2  T537E has few long microtubule-end binding events and the microtubule-end cannot accelerate mantADP dissociation.** (A) (B) Kymographs showing the interaction of WT MCAK and the T537E mutant (both green) with the microtubule (magenta). (C) (D) Histograms showing the end residence time of single molecules of WT MCAK and T537E. (E) Average fluorescence traces for the dissociation of mantADP from WT MCAK ($k = 0.102 \pm 0.013$ s$^{-1}$) and T537E ($k = 0.114 \pm 0.013$ s$^{-1}$) in solution and for T537E in the presence of 10 μM unpolymerised tubulin ($k = 0.120 \pm 0.019$ s$^{-1}$). (F) Average fluorescence traces for the dissociation of mantADP from WT MCAK ($k_1 = 4.11 \pm 0.41$ s$^{-1}$, $k_2 = 0.341 \pm 0.051$ s$^{-1}$) and T537E ($k = 0.369 \pm 0.089$ s$^{-1}$) in the presence of 5.7 μM microtubules.

the MCAK motor domain (Fig. 1A). Residues in the α4 helix are critical to MCAK's ability to recognise the microtubule end (*Patel et al., 2016*). This proximity to the α4 helix together with the observation that the mutation T537E specifically impairs the microtubule stimulated ATPase of MCAK suggests that this phosphomimic mutation may also interfere with MCAK's ability to distinguish the microtubule end from the lattice. We used TIRF microscopy to observe the interaction of single molecules of MCAK-GFP and T537E-GFP with microtubules. Wild type MCAK makes short diffusive interaction with the microtubule lattice. However, when it reaches the microtubule end ADP dissociation is accelerated, leading to nucleotide exchange and ATP. MCAK binds tightly and displays longer microtubule end residence events (Fig. 2A). The MCAK mutant T537E displays similar short diffusive interactions with the microtubule lattice but, by contrast with the

**Table 1** Compiled results of depolymerisation rate, ATPase activity, microtubule-end residence time and mantADP dissociation rates for wild type MCAK and T537E.

| MCAK variant | | WT | T537E |
|---|---|---|---|
| | | $(\mu m/min)$ (mean $\pm$ SD) | |
| **Depolymerisation rate** | | $3.04 \pm 0.53$ ($n = 20$) | $0.06 \pm 0.06$ ($n = 20$) |
| **ATPase activity** | | $(s^{-1})$ (mean $\pm$ SD) | |
| Solution (basal) | | $0.0045 \pm 0.0026$ ($n = 3$) | $0.0053 \pm 0.0033$ ($n = 3$) |
| Tubulin-stimulated | | $0.299 \pm 0.047$ ($n = 3$) | $0.194 \pm 0.055$ ($n = 3$) |
| fMicrotubule-stimulated | | $4.75 \pm 0.057$ ($n = 3$) | $0.335 \pm 0.081$ ($n = 3$) |
| | | $(s)$ (mean $\pm$ SEM) | |
| **Microtubule-end residence time** | | $2.03 \pm 0.13$ ($n = 289$) | $0.64 \pm 0.02$ ($n = 242$) |
| | | $(nM^{-1} s^{-1})$ (mean $\pm$ SD) | |
| **Microtubule-end interaction events** | | $0.71 \pm 0.29$ ($n = 15$) | $0.97 \pm 0.27$ ($n = 17$) |
| **mantADP dissociation**[a] | | $(s^{-1})$ (mean $\pm$ SD) | |
| Solution | | $0.102 \pm 0.013$ ($n = 3$) | $0.114 \pm 0.013$ ($n = 3$) |
| Tubulin-stimulated | | $0.114 \pm 0.023$ ($n = 3$) | $0.120 \pm 0.019$ ($n = 3$) |
| Microtubule-stimulated | First phase | $4.11 \pm 0.41$ ($n = 3$) | $0.369 \pm 0.089$ ($n = 3$) |
| | Second phase | $0.341 \pm 0.051$ ($n = 3$) | n/a |

Notes.
[a] The data and fits to the data from which these rate constants were obtained are shown in Fig. S2.

wild-type, does not show long end binding events (Fig. 2B). Quantification of microtubule end residence times for wild-type MCAK and T537E shows that 32% of molecules for the wild type but only 0.4% of molecules for T537E stayed at the microtubule end for longer than 2 s (Figs. 2C and 2D). The lattice residence time for T537E is unchanged relative to wild type ($0.42 \pm 0.01$ s and $0.48 \pm 0.02$ s, respectively) and neither the association or dissociation rates are significantly different (Fig. S1 and Table S1). These data indicate that the affinity for the microtubule lattice is not significantly changed for this mutant. By contrast, the end residence time for T537E was decreased to $0.64 \pm 0.02$ s compared to $2.03 \pm 0.13$ s for the wild type. In agreement with this, the rate of dissociation from the microtubule end is increased relative to wild-type (Table S1). To investigate the possibility that the mutation had impaired MCAK's ability to reach the microtubule end, we calculated the number of end interaction events per unit time for individual microtubules. This gave values of $0.71 \pm 0.29$ nM$^{-1}$ s$^{-1}$ and $0.97 \pm 0.27$ nM$^{-1}$ s$^{-1}$ for wild-type and T537E, respectively (Table 1). These data indicate that a similar number of MCAK molecules reach the microtubule end for both wild-type and mutant and that the phosphomimic mutation does not impair MCAK's ability to reach the microtubule end. Rather, in the

case of T537E, the average time a molecule remains at the microtubule end is significantly reduced. Taken together, these data imply that the molecular mechanism underlying the attenuation of depolymerisation activity due to this phosphomimic mutation is loss of ability to distinguish between the microtubule end and the microtubule lattice.

## The microtubule end does not accelerate ADP dissociation from T537E

ATP turnover by MCAK is maximally accelerated by the microtubule end due to microtubule end specific acceleration of ADP dissociation. To test whether the difference in T537E interaction with the microtubule end had affected the ability of the microtubule end to accelerate ADP dissociation, we measured the rate of dissociation of ADP tagged with the small fluorophore mant (mantADP). In solution and in the presence of unpolymerised tubulin, the rate constant for ADP dissociation was not significantly affected by this mutation (WT $0.102 \pm 0.013$ s$^{-1}$, T537E $0.114 \pm 0.013$ s$^{-1}$, WT + Tb $0.114 \pm 0.024$ s$^{-1}$, T537E + Tb $0.120 \pm 0.019$ s$^{-1}$) (Table 1 and Fig. 2E). This is in agreement with the ATPase activities under these conditions which are not significantly changed.

In the presence of microtubules, the change in fluorescence associated with mantADP dissociation from wild type MCAK is best described by a double exponential function. The two phases can be explained as a microtubule-end stimulated fast phase and a slower phase corresponding to molecules which do not encounter the microtubule end (*i.e.* remain in solution, in contact with unpolymerised tubulin or in contact with the microtubule lattice over the course of the experiment). The fast phase of mantADP dissociation from wild type MCAK measured here has a rate constant of $4.11 \pm 0.41$ s$^{-1}$ and the slower phase a rate constant of $0.341 \pm 0.051$ s$^{-1}$. The rate constant for the slower phase is in close agreement with the rate constant for mantADP dissociation from MCAK in solution or in the presence of unpolymerized tubulin. This is in agreement with this phase representing mantADP dissociation from molecules which do not contact the microtubule end. By contrast with WT-MCAK, the change in fluorescence associated with mantADP dissociation from T537E was well described by a single exponential function with a rate constant of $0.369 \pm 0.090$ s$^{-1}$. The faster phase was lost for the phosphomimic mutant indicating that the microtubule end cannot accelerate mantADP dissociation from T537E. In fact, the rate constant for the single phase observed for mantADP dissociation from T537E was not significantly different from the slower phase of mantADP dissociation from wild type MCAK ($p = 0.6679$, Fig. 2F and Table 1). Further, the microtubule-stimulated mADP dissociation rate constant is also similar to the rate constant in solution and in the presence of unpolymerized tubulin for both wild-type MCAK and T537E. Therefore, despite single molecule TIRF data showing that MCAK T537E can reach the microtubule end (Fig. 2B and Table 1), the kinetics of ADP dissociation for this mutant are consistent with the kinetics of ADP dissociation for wild-type MCAK when in solution, in the presence of unpolymerized tubulin or on the microtubule lattice. These data suggest that the phosphomimic mutant T537E has lost the ability to distinguish between the microtubule lattice and the microtubule end.

## DISCUSSION

To understand the molecular mechanism by which phosphorylation by Cdk1 impairs MCAK's depolymerisation activity, we studied the phosphomimic mutant T537E. This mutation causes a 50-fold decrease in the rate of microtubule depolymerisation by MCAK. Overexpression of this phosphomimic mutant in cells shows that it can localise to centromeres but that chromosome alignment is disrupted and cells arrest in metaphase (*Sanhaji et al., 2010*). The intra-centromere distance is decreased suggesting that cells expressing this mutant form of MCAK cannot generate tension across centromeres or correct erroneous chromosome attachments.

We have shown that, whilst the in solution and unpolymerized tubulin-stimulated ATPase activity of the phosphomimic mutant are unchanged, the microtubule stimulated ATPase is reduced 14-fold relative to wild-type MCAK. Single molecule TIRF analysis of this mutant showed that this impact on the microtubule-stimulated ATPase was due to loss of the ability of T537E to recognise the microtubule end. We saw that long end residence times, characteristic of wild-type MCAK's interaction with the microtubule, are abolished by the T537E mutation. Further, T537E cannot undergo microtubule-end stimulated acceleration of ADP dissociation, a crucial requirement for the depolymerisation activity of MCAK. Together our data show that T537E, unlike wild-type MCAK, cannot distinguish between the microtubule lattice and the microtubule end.

The localisation, abundance and activity of MCAK have all been shown to be regulated by phosphorylation at positions outside the motor domain by the kinases Aurora A and B, PLK1 and PAK1 (*Andrews et al., 2004*; *Ohi et al., 2004*; *Pakala et al., 2012*; *Sanhaji et al., 2014*; *Zhang, Ems-McClung & Walczak, 2008*). The MCAK neck, which is required for maximal microtubule depolymerisation activity (*Cooper et al., 2010*; *Ems-McClung et al., 2007*; *Ovechkina, Wagenbach & Wordeman, 2002*), is phosphorylated by both PAK1 and Aurora B kinase. Phosphorylation at this site inhibits depolymerisation activity (*Lan et al., 2004*; *Pakala et al., 2012*). This down regulation is likely due to disruption of an interaction between the neck-motor domain and C-terminal region required to allow MCAK to enter a high-affinity, depolymerisation competent conformation at the microtubule end (*Ems-McClung et al., 2013*). Phosphorylation of MCAK at its C-terminus by Aurora A has similarly been suggested to disrupt interactions between the neck-motor and C-terminal regions (*Talapatra, Harker & Welburn, 2015*).

The residue T537, the target of phosphorylation by Cdk1, is the only currently identified phosphorylation site within the MCAK motor domain. This residue is located adjacent to the C-terminal end of the α4 helix. The α4 helix has been suggested, on the basis of currently available structures of Kinesin-13 motor domains (*Asenjo et al., 2013*; *Ogawa et al., 2004*; *Shipley et al., 2004*; *Wang et al., 2017*), to play a role in deforming tubulin dimers thereby destabilising the microtubule and promoting depolymerisation. The ATP turnover cycle of MCAK differs from translocating kinesins and is adapted to promote tight binding of MCAK at the microtubule end where it can have maximal depolymerising impact (*Friel & Howard, 2011*). Previous work from our lab has shown that mutations in the α4 helix impact the molecular mechanism of MCAK in the same way as the phosphomimic mutant

described here (*Patel et al., 2016*). The proximity of T537 to the α4 helix may also provide the key to why the T537E mutant is unable to distinguish between the microtubule lattice and the microtubule end. The α4 helix is proposed to have a larger interface with tubulin at the intradimer groove in a curved conformation of tubulin, as may be found at the microtubule end, compared with a more constrained straight conformation within the microtubule lattice (*Asenjo et al., 2013*; *Ogawa et al., 2004*; *Wang et al., 2017*). Disruption of this crucial interface by mutation or by phosphorylation impairs MCAK's ability to distinguish different tubulin conformations and thereby recognise the microtubule end.

Our data shows that the T537E mutant, although significantly impaired in depolymerisation activity, can interact with the microtubule lattice with an affinity similar to wild-type MCAK. This mutant displays the characteristic diffusive interaction of MCAK with the microtubule lattice and can still reach the microtubule end. Assuming that the behaviour we have observed for this mutant is a good representation of the behaviour of phosphorylated MCAK's activity, MCAK phosphorylated at T537 is not prevented from interacting with microtubules of the mitotic spindle or being localised close to centromeres. However, it is locked in an inactive state that is blocked from recognising the microtubule end in the manner required to promote depolymerisation. Phosphorylation by Cdk1 holds MCAK in the set position; dephosphorylation is then the starter's gun allowing MCAK to instantly begin depolymerising kinetochore-attached microtubules at the correct moment. Phosphorylation at the Cdk1 site, T537, rather than completely blocking MCAK's interaction with microtubules, holds MCAK in an inactive state whilst allowing correct localisation. Thus, permitting the rapid switching of activity characteristic of regulation by phosphorylation.

## ACKNOWLEDGEMENTS

We are grateful to Alex Rathbone for technical support. Microscopy assays were carried out in the University of Nottingham School of Life Sciences Imaging (SLIM) facility. We thank the SLIM team and in particular Chris Gell for supporting our work.

### Funding

This work was supported by a BBSRC New Investigator award (BB/K006398/1) to Claire T. Friel, the Royal Society and the University of Nottingham. The funders had no role in study design, data collection and analysis, decision to publish, or preparation of the manuscript.

### Grant Disclosures

The following grant information was disclosed by the authors:
BBSRC New Investigator award: BB/K006398/1.
Royal Society.
University of Nottingham.

## Competing Interests

The authors declare there are no competing interests.

## Author Contributions

- Hannah R. Belsham performed the experiments, analyzed the data, wrote the paper, prepared figures and/or tables.
- Claire T. Friel conceived and designed the experiments, performed the experiments, analyzed the data, wrote the paper, prepared figures and/or tables.

## Data Availability

The raw data has been provided as Supplemental Files.

## Supplemental Information

Supplemental information for this article can be found online at http://dx.doi.org/10.7717/peerj.4034#supplemental-information.

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
