# Peer review of "A Cdk1 phosphomimic mutant of MCAK impairs microtubule end recognition"

_PeerJ, doi:10.7717/peerj.4034_

## Round 0.1 · original submission · Major Revisions

· Academic Editor

Major Revisions

Both reviewers appreciated the experimental design and the presentation of the data, however they both found that some of the conclusions in the manuscript need to be further substantiated or thoroughly revised. Please read their remarks on your proposed model carefully and address them. Both reviewers also raise a number of specific issues which you should also address.

·

Basic reporting

The microtubule depolymerizing kinesin-13, MCAK, plays critical roles in spindle assembly and in correcting improper kinetochore attachments. Because MCAK is a potent depolymerase, its activity is spatially and temporally regulated, but the mechanism of this regulation is not well understood. In the current paper the authors perform a thorough characterization of a T537 phosphomimic mutation of MCAK that lies in the catalytic domain of MCAK and which has been shown to be phosphorylated by Cdk1. They report that the T537E mutation results in decreased microtubule depolymerization activity due to its inability to remain at microtubule ends. This finding is a logical extension of their previous work in which they showed that mutation of highly conserved residues in alpha-helix 4 of MCAK also resulted in reduced microtubule depolymerization activity and a reduced binding to microtubule ends. Because T537 lies is positioned adjacent to alpha-helix 4, the current work provides strong corroborating evidence that this region is critical for recognition of microtubule ends and is an ideal way to modulate MCAK activity.

• The data presented is short, sweet, and to the point. The text is clearly written, and the presentation of the data makes it easy for the reader.
• The data is put in the context of the field, but the authors may have missed an opportunity to discuss the broader mechanisms of regulation of MCAK activity, discussed in more detail in point 3 below.
• It would be helpful to the reader if the authors reported what is plotted. For example, 1B represents a box and whiskers plot and 1C represents mean +/-???. The box and whisker definitions nor the error bar values (SD or SEM) are presented to the reader.
• For the original data supporting Fig. 1C, there is only one set of data for the +tubulin and +microtubule samples, and yet the Table says that n=3. Please clarify.
• In Fig. 2A, please provide the colors on the kymographs to the reader either on the figure or in the legend.
• For Fig. 2C and 2D, it would be helpful to the reader if the authors reported the fit of the data in the graph so that the reader does not have to go back and to the table. This is also true for Fig. S2. In addition, for each of these data sets, the excel sheet only has one set of data, and yet the tables say n=3. Since the data appear to be normalized, is it not possible to plot the average +/- SD of the normalized data in the graph? This would present all of the data to the reader.

Experimental design

• This is a well-done characterization of a mutant form of MCAK. The data provide a potential mechanistic understanding of how phosphorylation may regulate the catalytic function of MCAK.
• This is the first detailed biochemical characterization of a regulatory phosphorylation that occurs in the catalytic domain.
• All experiments appear technically sound with sufficient replicates.
• The methods are well-described.

Validity of the findings

• The results are clear, and the differences between wild type and mutant are clear and robust.
• The conclusions of the study are valid and well-justified.

Comments for the author

• In the models discussed at the end of the paper, the authors are not taking into account the multitude of observations showing that the other domains of MCAK target it to the kinetochore, inner centromere, spindle poles, or plus-tips of microtubules. Therefore, MCAK may be positioned near the microtubule end, so the key event is likely the residence time with the curved tubulin not just being at the end.
• It has also been well-established that MCAK’s microtubule depolymerization activity requires the positively charged neck region in addition to the catalytic domain. The neck is also the site of a critical Aurora A/B phosphorylation site, which has been shown to regulate microtubule depolymerization activity. Both the Wordeman Lab and the Walczak Lab have done extensive mechanistic analysis of how the neck and the regulatory phosphorylations affect MCAK activity. It would be helpful to the reader if the authors included some of this work in the discussion. For example, Ovechkina et al, 2002; Moore et al, 2004; Wagenbach et al, 2008; Cooper et al, 2010; Ems-McClung et al, 2013.

Reviewer 2 ·

Basic reporting

The authors describe the role of CDK1 mediated phosphorylation on the activity of MCAK. The background information provided in the text are well described and give a full picture of the biology of the kinesin MCAK.
The main conclusions of the work are that the T537E-MCAK phosphomimetic mutant has i) an attenuated microtubule depolymerisation activity; ii) a reduced ATPase rate in the presence of polymerised microtubules; iii) a shorter residence time at the microtubule end than the WT and iv) it cannot promote ADP dissociation.

Experimental design

The experiments are very well presented and designed. However, although the raw data support some of the conclusions, the general interpretation of the results need major revisions. Please find below the list of comments that the authors can consider during the revision of their work.

Validity of the findings

1. First set of results (described from line 99 to line 121). According to the data, the T537E-MCAK can turn over ATP in solution and bind to unpolymerised tubulin. How do the authors explain the differences in the ATPase rate of T537E-MCAK in the presence of unpolymerised vs. polymerized tubulin? Could the T537E mutation reduce the lattice-stimulated ATP cleavage due to a conformational change of T537E-MCAK which does not occur upon binding of unpolymerised tubulin?
The authors could measure the binding rate of T537E-MCAK to unpolymerised and polymerised tubulin and suggest some hypothesis on the mechanism of action of the MCAK mutant.

2. In the second paragraph of the results (lines 132-133) is written that “Wild type MCAK makes short diffusive interaction with the microtubule lattice”. It can be assumed that this is due to the initial high microtubule binding affinity of ATP.MCAK and, subsequently, to the lattice promoted ATP cleavage which weaken the MCAK interaction with the microtubules (Friel and Howard 2011, The EMBO Journal). Given the reduced ATPase rate of the MCAK phosphomimetic mutant in the presence of microtubules, it is very likely that T537E-MCAK is predominantly in the ATP-bound isoform. Thus, it is surprising that the affinity for the microtubule lattice is not significantly changed for MCAK mutant (line 138-139). Since ATP.MCAK tightly binds microtubule filaments, should not the authors expect a longer residency time for T537E-MCAK along the microtubule lattice and a reduced diffusion rate than the WT? For this reason, the authors might consider that the attenuation of the depolymerisation activity of the phosphomimetic mutant might be also due to the loss of ability of T537E-MCAK to reach the microtubules end (see also ‘General comments’ below).

3. In the third set of results (from line 148 to line 176) is shown that the T537E-MCAK mutant has a slow ADP dissociation rate at microtubules end. However, the rationale and the conclusions of this experiment (see also ‘General comments’ below) do not seem very clear and robust.
Concerning the rationale of the experiment, I am wondering whether the number of T537E-MCAK molecules bound to ADP at the microtubule end is sufficient to test the ADP dissociation rate. The data presented in paragraph 1 and 2 of the results show that it is very unlikely that T537E-ADP.MCAK can reach the microtubule end because of i) the reduced ATPase rate of T537E-MCAK bound to microtubules (0.335±0.081s-1 compared to 4.75±0.057s-1 in WT) and ii) the short residence time of T537E-MCAK at the microtubule end (0.64±0.02s compared to 2.03±0.13s for the WT). For these reasons, I suggest moving these experiments in the supplementary data section because they are supporting previously results but do not significantly add novel information.

Comments for the author

a. At lines 213-215 the authors write that “This mutant still displays the characteristic diffusive interaction of MCAK with the microtubule lattice and can still reach the microtubule end”. However, this statement is not very well supported in the text/figures in which, instead, is shown that T537E-MCAK has a reduced residence time at the microtubules end. In addition, this sentence does not sustain the conclusion that T537E-MCAK does not distinguish between the microtubule lattice and the microtubule end. Although this conclusion might be very intriguing to explain the attenuated T537E-MCAK depolymerisation activity, there is no direct proof for this and sounds too speculative at this stage of the work.

b. The conclusion that the reduced depolymerisation rate of T537E-MCAK is dependent on the attenuated ADP dissociation from T537E-MCAK at the microtubules end, might not be fully appropriate (see also point 3 above). Indeed, the authors do not consider the hypothesis that, given the reduced residence time at the ends, the mutant MCAK that does not get to the microtubule tip because is not able to diffuse along the lattice (see also point 2 above). This hypothesis is also in accordance with another work showing that T537E-MCAK predominantly localizes along the microtubule lattice of the mitotic spindle (but not at the ends) and display a very weak staining at centromeres in HeLa cells (Sanhaji et al., 2010, Mol Cell Biol).
In other words, the authors might discuss and consider the hypothesis that the CDK1 mediated phosphorylation of MCAK does not directly affect the catalytic activity of the kinesins at microtubule end but regulates the ability to diffuse along the microtubule lattice by modulating the ATP-ADP cycle on MCAK.

---

## Round 0.2 · accepted · Accept

· Academic Editor

Accept

You will see that the Reviewer has appreciated the changes that you have introduced in your revised version, both in terms of experimental additions and in terms of improved clarity.

Reviewer 2 ·

Basic reporting

The authors have addressed all points listed during the first revision of the paper. The results are very well explained and the conclusions are robust.

Experimental design

I appreciate the authors have now performed the experiment to investigate the ability of the T537E-MCAK mutant to get to the microtubule end. This additional experiment clearly show the T537E-MCAK phosphomimetic mutant can reach the microtubule end although its residence time is much shorter than the wild type. Overall, this experiment greatly contributes to support the conclusion that the T537E-MCAK is not able to distinguish between microtubule lattice and microtubule end.

Validity of the findings

The findings about the role of the CDK1-mediated phosphorylation of MCAK are strongly supported by the data and I recommend this work to be published in PeerJ.